# Enhanced local feature extraction of lite network with scale-invariant CNN for precise segmentation of small brain tumors in MRI

Wei Yuan[1], Han Kang [ID][2]*

**1** College of Computer Science, Chengdu University, Chengdu, China, **2** Prenatal diagnosis department, Chengdu Women's and Children's Central Hospital, School of Medicine, University of Electronic Science and Technology of China, Chengdu, China

* 312208880@qq.com

**Data availability statement:** All relevant data are within the manuscript and its Supporting information files.

## Abstract

Deep learning has emerged as the preeminent technique for semantic segmentation of brain MRI tumors. However, existing methods often rely on hierarchical downsampling to generate multi-scale feature maps, effectively capturing fine-grained global features but struggling with large-scale local features due to insufficient network depth. This limitation is particularly detrimental for segmenting diminutive targets such as brain tumors, where local feature extraction is crucial. Augmenting network depth to address this issue leads to excessive parameter counts, incompatible with resource-constrained devices. To tackle this challenge, we propose that object recognition should exhibit scale invariance, so we introduce a shared CNN network architecture for image encoding. The input MRI image is directly downsampled into three scales, with a shared 10-layer convolutional network employed across all scales to extract features. This approach enhances the network's ability to capture large-scale local features without increasing the total parameter count. Further, we utilize a Transformer on the smallest scale to extract global features. The decoding stage follows the UNet structure, incorporating incremental upsampling and feature fusion from previous scales. Comparative experiments on the LGG Segmentation Dataset and BraTS21 dataset demonstrate that our proposed LiteMRINet achieves higher segmentation accuracy while significantly reducing parameter count. This makes our approach particularly advantageous for devices with limited memory resources. Our code is available at https://github.com/chinaericy/MRINet.

## Introduction

Brain tumors are abnormal growths of cancerous or non-cancerous cells within the brain, it is classified based on the type and severity of benign and malignant tumors [1–3].They can be either benign or malignant. Benign brain tumors have a uniform structure and do not contain active (cancerous) cells, while malignant brain tumors have an irregular structure and contain active cells. Examples of low-grade tumors include gliomas and meningiomas,

**Funding:** The author(s) received no specific funding for this work.

**Competing interests:** The authors have declared that no competing interests exist.

which are classified as benign tumors. They resemble normal brain cells and grow slowly. On the other hand, glioblastomas and astrocytomas are examples of high-grade tumors, classified as malignant tumors [4]. They typically grow rapidly.In 2018, the 5-year relative survival rate for patients diagnosed with primary malignant brain and other central nervous system tumors was only 34.9%, whereas for cases of primary non-malignant brain tumors, this rate was 90.47% [5]. Therefore, effective segmentation of brain tumor images to understand details and provide appropriate treatment is crucial. Mathematical algorithms can be used for feature extraction, modeling, and measurement in MRI images to detect pathological changes, disease progression, or to compare normal subjects with abnormal subjects [6]. Accurate and reproducible quantification of the size and morphology of brain tumors is crucial for diagnosis, treatment planning, and monitoring the tumor's response to therapy [7].

Ali et al. [8] conducted a review on the current status of MRI brain tumor segmentation using deep learning techniques. They elucidated the remarkable achievements of these deep learning methods in effectively processing and evaluating large input image data compared to traditional approaches.Three datasets re developed consisting of MRI sequences by Tandel et al. [9] to optimize the classification capacity bet en low-grade and glioma. In addition, some widely used convolutional neural networks reused for tumour classification: AlexNet [10], VGG16 [11], GoogleNet [12], and ResNet50 [13]. In order to get more uniform and superior outcomes than any single DL model, it was proposed to use the mainstream DL models in an ensemble method.

UNet [14], as a widely used model, has achieved significant success in the field of medical image segmentation. It is derived from the fully convolutional network (FCN) [15] and is named after its overall structure resembling the letter "U". The UNet architecture consists of an encoder and a decoder, which extract features through symmetric down-sampling and up-sampling paths. The encoder progressively reduces the image size to extract deep-level features using convolutional layers and pooling layers, while the decoder gradually restores the spatial resolution of the input image through deconvolution. Fabian et al. [16] demonstrated the effectiveness of a well-trained UNet in the BraTS 2018 challenge, where they were able to improve segmentation performance with a slight tweak to the network design. They used a large patch size to create the training dataset and trained the network on that dataset using the dice loss function. They achieved high Dice scores on the validation data and won the second place in the competition. Dong et al. [17] proposed a fully automatic brain tumor segmentation method using a UNet-based deep convolutional network, which showed strong performance in the core region and competitive results in the whole tumour region. Havaei et al. [18] proposed a fully automated brain tumour segmentation technique specifically for low grade and high grade glioblastoma MRI images. They introduced a novel CNN architecture with fully connected layers, resulting in a 40-fold speedup. In addition, they implemented a two-stage training process to cope with the challenge of tumour label imbalance. In addition, many researchers have extensively studied the UNet model with various modifications.Zhou et al. [19] proposed UNet++, which aims to enhance the semantic consistency between the encoder and decoder subnetwork feature maps by combining the encoder-related feature maps with the decoder's feature maps before combining the encoder's feature maps with the decoder's feature maps. This approach aims to narrow the semantic gap between the feature maps of the encoder and decoder sub-networks, thus improving the segmentation efficiency. Xiao et al. [20] added the Squeeze Excitation Residual (SER) module and the Atrous Spatial Pyramid Pooling (ASPP) module to UNet++, which are respectively aimed at alleviating the gradient vanishing problem and obtaining contextual information. They proposed a novel SAUNet++ model and used it to segment COVID-19 lesion images. The experimental results show that the segmentation performance of SAUNet++ outperforms

that of UNet++ in various metrics. In addition to UNet and its variants, many other models have been used for segmentation tasks, including FNN [21], SegNet [22], RefineNet [23], and so on.

However, CNNs have their limitations, such as slow parameter updates during back-propagation, convergence to local optima, information loss in pooling layers, and unclear interpretation of feature extraction. The Transformer [24] model was initially introduced by the Google team in 2017, replacing the convolutional neural network components with self-attention modules. This model utilizes multiple attention heads to process and capture different input data features, thereby enhancing feature extraction capabilities. Ben Graham et al. [25] proposed a hybrid neural network named LeViT, which is a Vision Transformer-based model that can improve inference speed while maintaining accuracy. Xu et al. [26] introduced LeViT-UNet, which integrates LeViT modules into the UNet architecture for fast and accurate medical image segmentation. Li et al. [27] introduced MA-UNet, an improved segmentation network built upon UNet. It integrates multi-scale features and incorporates a hybrid attention mechanism. Hsienchih et al. [28] proposed a brain tumor segmentation method based on a multi-modal transformer, utilizing the UNet architecture. Firstly, the convolutional encoder extracts features from each modality. Subsequently, the multi-modal transformer model establishes correlations between multi-modal features and learns features from missing modalities. Finally, the multi-modal shared-weight decoder utilizes spatial and channel self-attention modules to gradually aggregate multi-modal and multi-level features. This includes modality-specific encoders, a multi-modal transformer, and a multi-modal shared-weight decoder, all employed to accomplish the brain tumor segmentation task. Mariia et al. [29] combined CNN with Transformer for tumor image feature extraction. Their proposed method achieved a Dice score of 0.9256 and an HD95 score of 3.374 on the validation set of BraTS 2021. However, Transformers process images block by block, leading to excessive computational overhead. To address this issue, the Microsoft team introduced Swin Transformer [30]. Unlike the traditional Transformer, Swin Transformer divides the image into multiple uniformly sized windows and restricts Transformer computations to within each window to reduce computational complexity. Meshram et al. [31] utilized Swin Transformer for brain tumor detection, and experimental results showed that compared to Transformer, Swin Transformers required less model training time and provided better computational metrics. Venkata et al. [32] proposed an improved Lite Swin Transformer for brain MRI segmentation, using open-source brain tumor images from the Kaggle database. Their proposed deep learning method was tested against other well-known transfer learning methods on the same MRI dataset. The results indicated that although the processing time of the images increased, the segmentation results were improved.

Although there are many CNN-based networks, Transformer networks, or hybrid networks for semantic segmentation of brain MRI tumors, these networks either directly use small-scale global features captured by deep networks for decoding, or fuse small-scale global features captured by deep networks with local features captured by shallow networks for decoding. Some networks increase the depth of each scale, resulting in an increase in network parameters. This paper addresses the contradiction between using deep large-scale networks to capture local features and the number of network parameters. The main innovations are as follows:

1) We proposed that object recognition should exhibit scale invariance. Based on this principle, we introduced a structure with multiple large-scale shared CNNs, allowing the network to capture finer large-scale local features without increasing the network parameters.

2) We proposed a super lite network model named LiteMRINet, that allows multiple large-scale shared CNNs to capture finer local features and Transformer to capture global features .

With only few parameters, the network can perform well in the small target segmentation task like tumor.

3) We conducted comparative experiments on various mainstream networks on a semantic segmentation dataset of brain MRI tumors, analyzing their accuracy and parameter count.

## Materials and methods

### Methods

For semantic segmentation of medical MRI images, the UNet architecture is considered a landmark presence. UNet and many subsequent semantic segmentation networks adopt a progressive downsampling approach to increase receptive fields and extract deeper features. Typically, large-scale feature maps contain rich global feature information, while small-scale feature maps primarily consist of local features. Small-scale feature maps, due to their smaller size, allow convolutional operations of the same size to have a larger receptive field, thus having better global feature capturing capabilities. However, due to downsampling operations, many detailed information is lost, resulting in poor performance in capturing local features. Therefore, semantic segmentation networks like UNet employ a progressive fusion of large-scale features during decoding to merge small-scale global features with large-scale local features. See Fig 1 for illustration.

Although networks like UNet, which employ progressive downsampling, can to some extent capture both global and local features and fuse them, the large-scale local features are solely extracted by shallow layers, and the network's depth is insufficient to fit local features

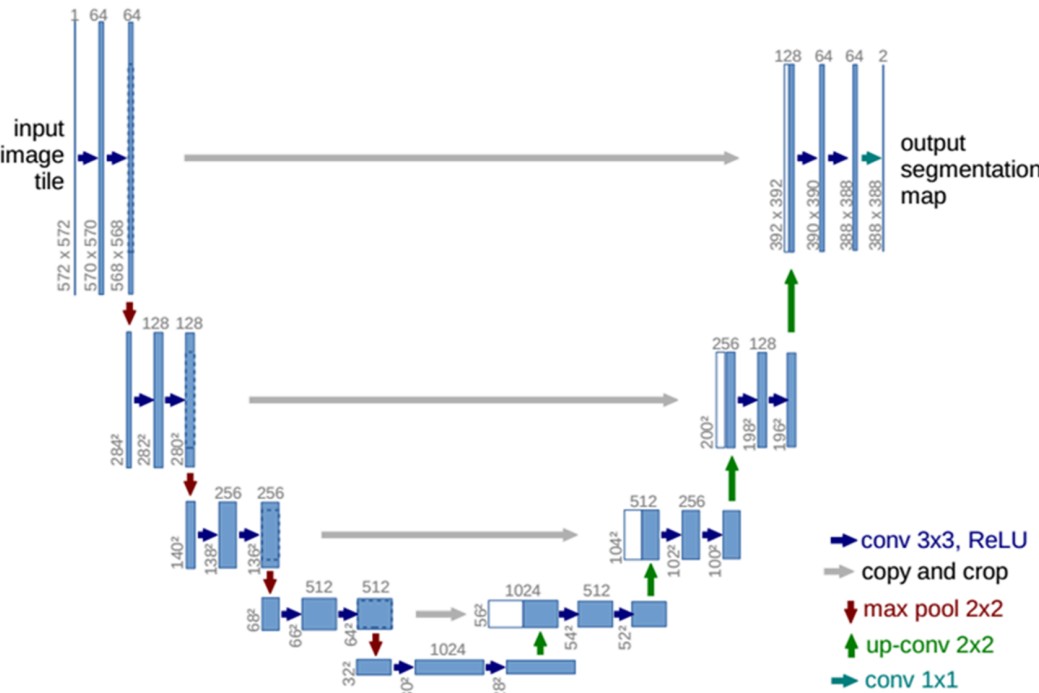

**Fig 1. UNet architecture (example for 32x32 pixels in the lowest resolution).** Each blue box corresponds to a multi-channel feature map. The number of channels is denoted on top of the box. The x-y-size is provided at the lower left edge of the box. White boxes represent copied feature maps. The arrows denote the different operations.

well, resulting in very limited local feature extraction capability. Moreover, in brain tumor segmentation on MRI images, there is typically only one tumor, and it often occupies a small proportion of the image, making it a small target segmentation problem. The accuracy of such small target segmentation heavily relies on the quality of local features. However, if we increase the depth of the large-scale network, the number of parameters in the network will increase significantly.

We all know that convolution exhibits translation invariance, which means that the convolution operation can detect an object regardless of its position in the image. In other words, the convolution operation is independent of the object's location within the image. Inspired by the translation invariance of convolution, we hypothesize that object recognition should also be independent of the scale of the image, suggesting that convolutional kernels should maintain scale invariance. As illustrated in Fig 2, regardless of the variations in the input image size, the convolutional network should be able to detect the green tumor within the image.

Eq (1) represents the expression for convolution. In it, $H$ denotes the convolution output, with subscript $s$ indicating scale, $i$ representing the horizontal coordinate of a pixel, $j$ representing the vertical coordinate of a pixel, $\Delta$ denoting the size of the convolution kernel, $V$ representing the weights that exhibit translation invariance, $X$ representing the input pixel values, $a$ indicating the horizontal offset, and $b$ indicating the vertical offset. For UNet, each scale of feature extraction has its own independent convolution kernel; for example, five scales would have five independent convolution kernels. As shown in Fig 2, we believe that regardless of the changes in image size, convolution should be able to detect the target, meaning that the convolutional layer should possess scale invariance. Therefore, we can remove the subscript $s$ from the convolution kernel $V$ in Eq (1), as shown in Eq (2). In Eq (2), inputs $X$ at different

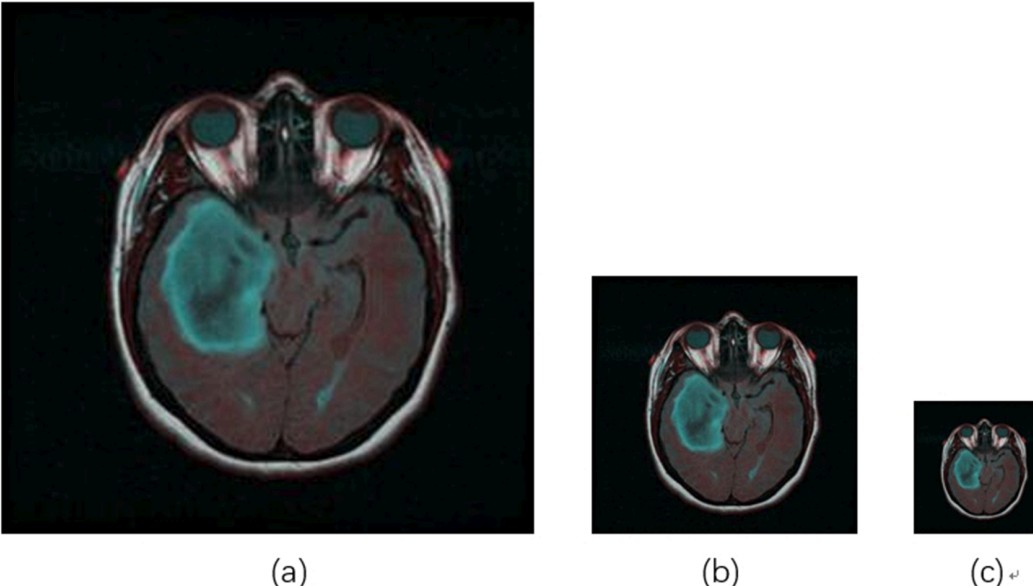

**Fig 2. Different scales of the same image.** (b) is half the size of (a), and (c) is half the size of (b).

scales share the same convolution kernel *V*.

$$[H]_{s,i,j} = u + \sum_{a=-\Delta}^{\Delta} \sum_{b=-\Delta}^{\Delta} [V]_{s,a,b}[X]_{s,i+a,j+b} \tag{1}$$

$$[H]_{s,i,j} = u + \sum_{a=-\Delta}^{\Delta} \sum_{b=-\Delta}^{\Delta} [V]_{a,b}[X]_{s,i+a,j+b} \tag{2}$$

To address the contradiction between the depth of the network on a large scale and the total number of network parameters, I propose a network structure called LiteMRINet. LiteMRINet adopts a multi-scale shared network, where CNN layers are shared across three large scales, increasing the depth of the network at each scale to better capture local features. Additionally, to enhance the capture of global features, Transformer is used instead of convolution operations on a small scale. Since the feature map size processed by the network at the small scale is relatively small, the parameters of the Transformer will not be too many, and replacing convolution with Transformer will not significantly increase the number of parameters and computational complexity of the network. The network architecture is illustrated in Fig 3.

In Fig 3, the input MRI image has dimensions of 256×256 with 3 channels. Then, the image is downsampled to 128×128×3 and 64×64×3, respectively. Subsequently, the MRI images at these three scales undergo 10 layers of CNN operations with parameter sharing to extract deep-level local features. Each CNN layer consists of 2D convolutions with a kernel size of 3×3, a stride of 1, BatchNorm, and ReLU activation. At this stage, local feature maps of sizes 256×256×64, 128×128×64, and 64×64×64 are obtained for the three scales. Next, the feature map of size 64×64×64 is max-pooled with a stride and kernel size of 2, resulting in a

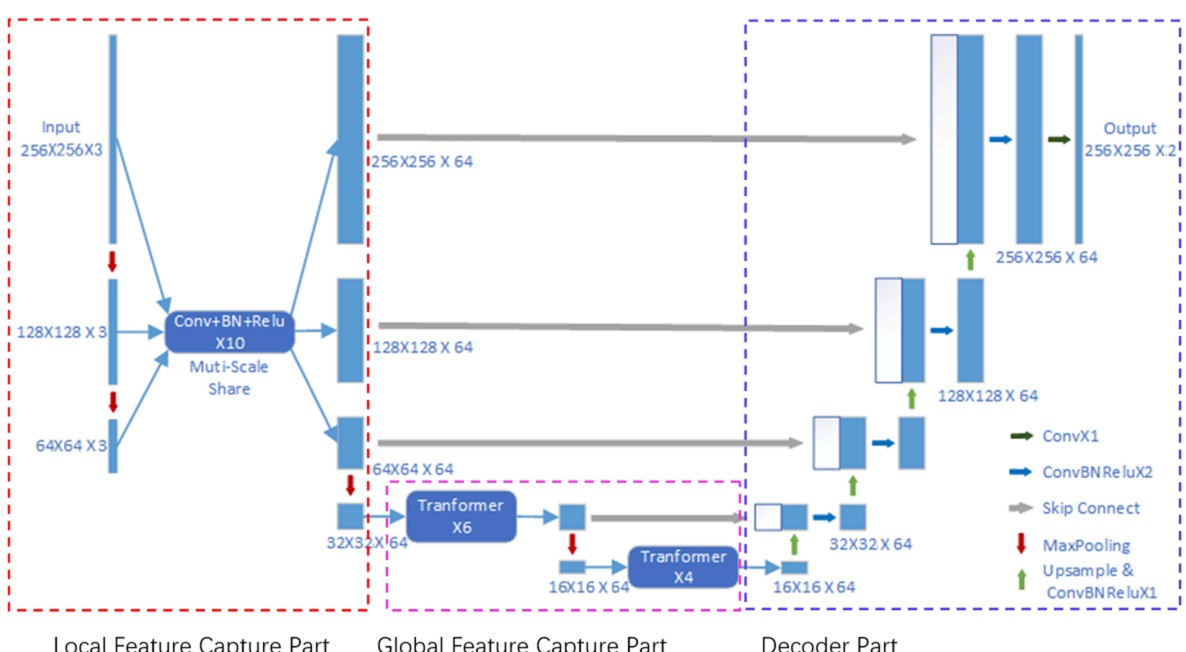

**Fig 3. Architecture diagram of LiteMRINet.**

small-scale feature map of size 32×32×64. After passing through a global feature extraction layer composed of 6 Transformers, a global feature map of size 32×32×64 is obtained. Subsequently, the 32×32×64 global feature map undergoes max-pooling again, with a stride and kernel size of 2, yielding a small-scale feature map of size 16×16×64. Finally, after passing through a global feature extraction layer composed of 4 Transformers, a global feature map of size 16×16×64 is obtained. In the global feature extraction layers, each Transformer utilizes a multi-head attention mechanism with a head number of 8.

The decoding part follows the same structure as UNet. Initially, the 16×16×64 feature map is upsampled by a factor of two, followed by a single CNN layer operation, resulting in a size of 32×32×64. After concatenating with the feature map of size 32×32×64, two additional CNN layers are applied, yielding a decoded map of size 32×32×64. Each CNN layer consists of 2D convolutions with a kernel size of 3×3, a stride of 1, BatchNorm, and ReLU activation. This process is repeated in a similar manner for each subsequent upsample, until a decoded map of size 256×256×64 is obtained. Finally, to output the tumor mask, a single 2D convolution operation with a kernel size of 1 and stride of 1 is applied to adjust the channel dimension of the decoded map to 2.

## Materials

a) LGG Segmentation Dataset [33]

LGG segmentation dataset contains brain MR images together with manual FLAIR abnormality segmentation masks.The images were obtained from The Cancer Imaging Archive (TCIA). They correspond to 110 patients included in The Cancer Genome Atlas (TCGA) lower-grade glioma collection with at least fluid-attenuated inversion recovery (FLAIR) sequence and genomic cluster data available.

We compiled the MRI images and their corresponding masks for these 10 patients, resulting in a total of 3929 images and their respective masks. Out of these, 3000 images were allocated for the training set, while the remaining 929 images were designated for the test set. Considering the limitations of machine memory, we resized all images proportionally to a size of 256×256. In the masks, pixels representing non-tumor areas were set to 0, while pixels indicating tumor regions were set to 1.

b) BraTS21 dataset

The BraTS21 dataset was derived from the RSNA-MICCAI Brain Tumor Radiogenomic Classification Competition data. Specifically, the training images from the competition were preprocessed using Torchio to ensure they all had an Axial view. Then, the selected slices of FLAIR, T2w, and T1wCE were combined into a three-channel image. Finally, simple thresholding, median filtering, and morphological operations were applied to the FLAIR MRI for tumor segmentation, resulting in the corresponding mask.The training set contains 580 images and the test set includes 85 images.

The tumor segmentation masks are 2D numpy arrays; therefore, we converted them into single-channel PNG images, where the region with pixel values of 1 represents the tumor area, and the rest of the pixels are 0. The dataset images have a size of 224 x 224, and to align with the LGG Segmentation Dataset, both the images and masks were resized to 256 x 256.

## Results

### Experimental environment

The hardware configuration of the computer used for the experiments includes Intel i5-13600KF CPU, SEIWHALE DDR4 16G × 3 RAM, and NVIDIA GeForce RTX 4060TI 16G

GPU. The Python version is 3.6.8, and PyTorch is used as the deep learning framework for model training and evaluation. Backpropagation was performed using AdamOptimizer [34] with a batch size of 4 and a learning rate of 0.0001. The default value of EPS was adjusted to 0.003 as it is too small and may lead to NAN loss during training for some models. The total loss consists of L2 regularisation and binary cross-entropy, which is used to prevent overfitting as shown in Eq. 1. The maximum number of training epochs was set to 300, and each epoch was evaluated on the validation dataset afterwards. Unlike the stopping criterion used in Shift Pooling PSPNet [35], our stopping criterion is to stop training if the metrics of the test dataset do not improve in 10 consecutive epochs, and to stop training if the loss of the test dataset does not decrease in 20 consecutive epochs, provided that the total number of training epochs exceeds 100.

$$\left.\begin{array}{l} \text{TotalLoss} = \text{BinaryCrossEntropy} + L2 \\ L2 = \|w\|_2^2 = \sum_i |w_i^2| \end{array}\right\} \tag{3}$$

## Evaluation metrics

Because precision and recall have a trade-off relationship, making direct comparisons challenging, we employed F1-Score, mIoU (Mean Intersection over Union), and OA (Overall Accuracy) to evaluate each deep learning model. They are calculated as shown in Eqs (2)–(6).

The formula of mIoU is:

$$\text{mIoU} = \frac{1}{N+1} \sum_{i=0}^{N} \frac{TP}{TP + FN + FP} \tag{4}$$

The formula of OA is:

$$\text{Accuracy} = \frac{TP + TN}{TP + TN + FP + FN} \tag{5}$$

The formula of the F1-score is:

$$\text{F1-Score} = \frac{2 \times Precison \times Recall}{Precision + Recall} \tag{6}$$

where precision and recall are:

$$Precison = \frac{TP}{TP + FP} \tag{7}$$

$$Recall = \frac{TP}{TP + FN} \tag{8}$$

In formula (2), N is the number of the foreground. TP is the abbreviation of true positives, that is, the number of pixels correctly predicted as the foreground. FP is the abbreviation of false positives, that is, the number of background pixels misjudged as the foreground. TN is the abbreviation of true negatives, that is, the number of pixels correctly predicted as the background. FN is the abbreviation of false negatives, that is, the number of foreground pixels misjudged as background. Tumor and background are regarded as the foreground to obtain IoU, then the average value is taken as mIoU. The foreground in Eqs (2)–(4) is the tumor.

## Experimental results on LGG segmentation dataset

To objectively compare the performance of our proposed LiteMRINet with other network models, we conducted comparative experiments on the LGG Segmentation Dataset. First, we

trained UNet, PSPNet, SegNet, Attention UNet, NestedUNet, LeViT Net and LiteMRINet on the training set. Then, we evaluated their performance on the test set, calculating the values of mIoU, F1-score, and OA, three objective evaluation metrics. The results are presented in Table 1.

As can be seen from Table 1, these models can be roughly divided into three groups. The first group comprises models with relatively low accuracy, including LeViT UNet and Seg-Net. Within this group, LeViT UNet performs the worst, achieving mIoU of 80.2, an F1-Score of 75.65, and an OA of 99.6. SegNet performs slightly better, with an mIoU of 81.75, an F1-Score of 77.95, and the same OA of 99.6. However, overall, the mIoU and F1-Score of SegNet remain very low.

PSPNet, which uses ResNet50 as its backbone, falls into the group with intermediate performance metrics: it achieves an mIoU of 85.1, an F1-Score of 82.7, and an OA of 99.7. UNet, a classic medical image segmentation model, delivers moderate performance, ranking exactly in the middle among all seven models, with an mIoU of 86.25, an F1-Score of 84.2, and an OA of 99.7. Notably, Nested UNet which is an improved version of the UNet architecture, does not perform better on this dataset; all its metrics are identical to those of UNet. This indicates that the Nested UNet algorithm does not exhibit its generalization advantage on datasets with low annotation accuracy.

Attention UNet and the proposed LiteMRINet (in this study) have relatively close performance metrics, together forming the group with superior accuracy. Within this group, Attention UNet achieves an mIoU of 87.0, an F1-Score of 85.2, and an OA of 99.7. The proposed LiteMRINet outperforms all other models, with an mIoU of 87.2, an F1-Score of 85.5, and an OA of 99.7. Compared with LeViT UNet (the worst-performing model), LiteMRINet's mIoU and F1-Score are 7.0 and 9.85 percentage points higher, respectively.

Fig 4 illustrates the tumor prediction images of each model on a brain MRI from the test set. In the first row, the first image on the far left is the original brain MRI, the second image is the ground truth label, and the third image shows the prediction result of the UNet model. In the second row, from left to right, the prediction results of PSPNet, SegNet, and Attention UNet are displayed. The third row shows the prediction results of Nested UNet, LeViT UNet, and our proposed LiteMRINet, respectively, from left to right. In the image, red pixels represent false negatives, indicating tumor pixels in the original image that were missed in the prediction. Green pixels represent false positives, indicating non-tumor pixels in the original image that were mistakenly identified as tumors. Black pixels represent true negatives, indicating pixels correctly predicted as non-tumor. White pixels represent true positives, indicating tumor pixels correctly identified. In these prediction result images, all models can roughly detect tumor pixels, but there are differences in accuracy, this is particularly evident in the performance on edge pixels. This phenomenon underscores the importance of local features in the task of brain MRI tumor segmentation.

**Table 1. Results of classic semantic segmentation on the test dataset.**

| Methods | mIoU (%) | F1-Score (%) | OA (%) |
|---|---|---|---|
| LeViT UNet | 80.2 ± 0.2 | 75.65 ± 0.25 | 99.6 ± 0 |
| SegNet | 81.75 ± 0.15 | 77.95 ± 0.25 | 99.6 ± 0 |
| PSPNet | 85.1 ± 0.1 | 82.7 ± 0.2 | 99.7 ± 0 |
| UNet | 86.25 ± 0.05 | 84.2 ± 0.1 | 99.7 ± 0 |
| Nested UNet | 86.25 ± 0.05 | 84.2 ± 0.1 | 99.7 ± 0 |
| Attention UNet | 87.0 ± 0.2 | 85.2 ± 0.2 | 99.7 ± 0 |
| LiteMRINet | 87.2 ± 0.2 | 85.5 ± 0.2 | 99.7 ± 0 |

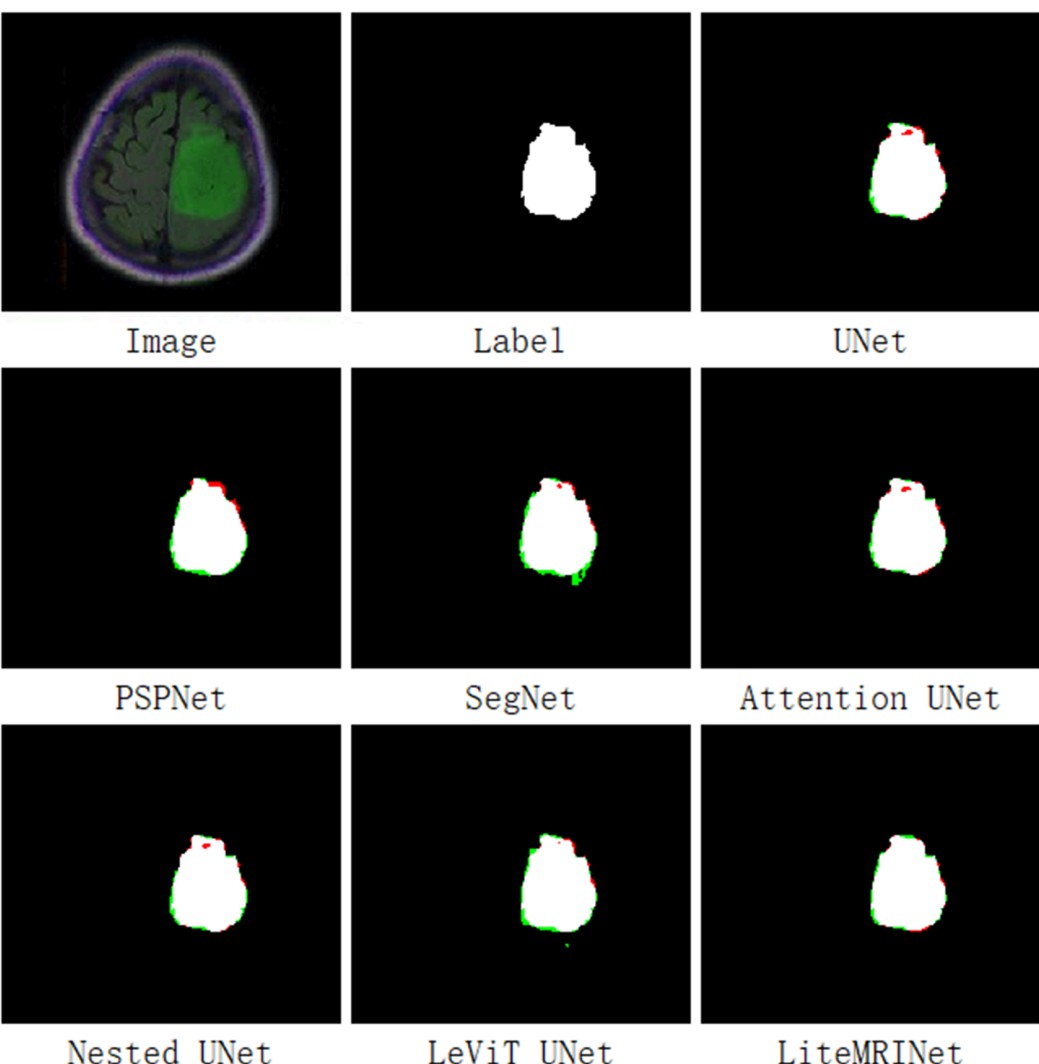

**Fig 4. The output results of each model on the test set.** Black represents TN pixels, white represents TP pixels, red represents FN pixels, and green represents FP pixels.

UNet, PSPNet, Attention UNet, and Nested UNet exhibit a large number of red pixels at the edges, indicating significant missed detections. In contrast, LeViT UNet and SegNet have more green pixels; the green pixels at the lower edge of SegNet are particularly prominent. This suggests that these two models have a high number of false detections. The proposed LiteMRINet model stands out: it not only has fewer missed detection pixels but also an extremely small number of false detection pixels, with much more accurate edges compared to the other models.

Fig 5 presents the tumor prediction results of various models on another brain MRI image from the test set. There is an unknown black hole region in the middle of this MRI image. For such black hole regions, the SegNet and LeViT UNet models exhibit an extremely high missed detection rate, which is reflected by the large number of red pixels in their prediction images. UNet, PSPNet, and Nested UNet perform slightly better, yet their prediction images still contain a significant number of red pixels that indicate missed detections. Attention Net and the

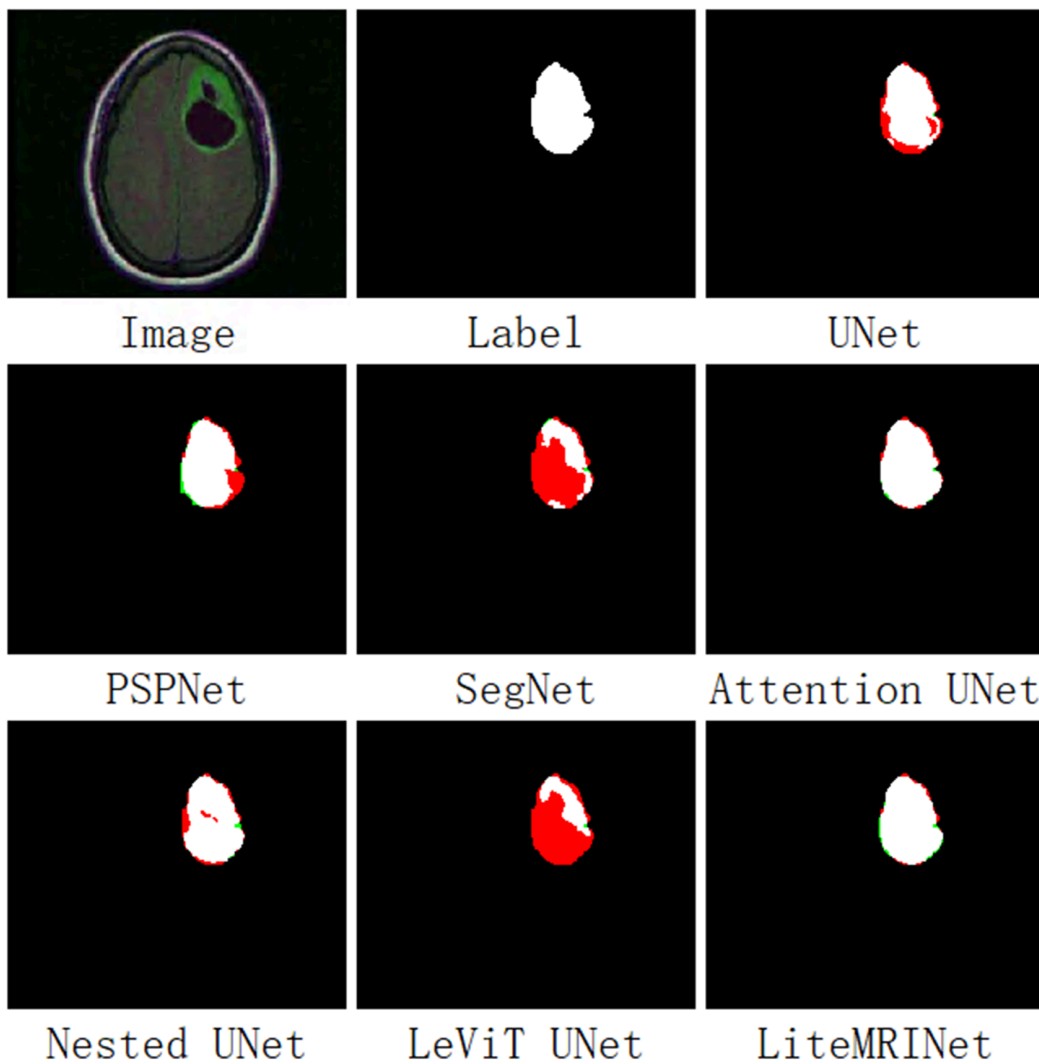

**Fig 5. The output results of various models on MRI images with tumor holes in the test set.** Black represents TN pixels, white represents TP pixels, red represents FN pixels, and green represents FP pixels.

proposed LiteMRINet achieve the lowest missed detection rates: there are essentially no red pixels at the tumor-associated black hole regions, with only a minimal amount of red pixels appearing at the upper edges of the tumors. This result demonstrates that the proposed large-scale shared convolutional network, compared with other models, possesses superior local feature-capturing capabilities and yields lower error rates in the task of tumor edge segmentation.

## Experimental results on BraTS2021 dataset

To further validate the effectiveness of our proposed LiteMRINet, we conducted comparative experiments on the BraTS2021 Dataset. The results are presented in Table 2.

As can be seen from Table 2, the performance of the various models on the test dataset can be divided into two groups.

**Table 2. Results of classic semantic segmentation on the test dataset.**

| Methods | mIoU (%) | F1-Score (%) | OA (%) |
| --- | --- | --- | --- |
| UNet | 87.4 ± 0.2 | 86.7 ± 0.2 | 98.4 ± 0 |
| Nested UNet | 87.45 ± 0.15 | 86.8 ± 0.2 | 98.4 ± 0 |
| Attention UNet | 87.5 ± 0.2 | 86.8 ± 0.2 | 98.4 ± 0 |
| SegNet | 87.65 ± 0.05 | 86.95 ± 0.05 | 98.4 ± 0 |
| PSPNet | 89.9 ± 0.2 | 89.6 ± 0.2 | 98.75 ± 0.05 |
| LeViT UNet | 90.85 ± 0.05 | 90.65 ± 0.05 | 98.9 ± 0 |
| LiteMRINet | 91.25 ± 0.25 | 91.05 ± 0.25 | 98.9 ± 0 |

The first group consists of four UNet variants: UNet, Nested UNet, Attention UNet, and SegNet. These models exhibit relatively low and highly similar accuracy, with no significant differences among them. Among these, UNet performs the weakest, achieving an mIoU of 87.4, an F1-Score of 86.7, and an OA of 98.4. Nested UNet performs almost identically to UNet, with an mIoU of 87.45, an F1-Score of 86.8, and an OA of 98.4. Attention UNet also shows no obvious difference from UNet, delivering an mIoU of 87.5fig5 an F1-Score of 86.8, and an OA of 98.4. While SegNet performs slightly better, the improvement is also insignificant, it achieves an mIoU of 87.65, an F1-Score of 86.95, and an OA of 98.4.

The second group, comprising PSPNet, LeViT UNet, and LiteMRINet, demonstrates better performance. PSPNet achieves an mIoU of 89.9, an F1-Score of 89.6, and an OA of 98.75, which is significantly superior to SegNet. LeViT UNet further improves these metrics to 90.85 (mIoU), 90.65 (F1-Score), and 98.9 (OA). Among all models, the proposed LiteMRINet performs the best, with an mIoU of 91.25, an F1-Score of 91.05, and an OA of 98.9. Although the OA values of all models are very close, LiteMRINet stands out with significantly better mIoU and F1-Score, which fully demonstrates its superior effectiveness.

Fig 6 presents the tumor prediction performance of different models on brain MRI images from another dataset, BraTS2021. PSPNet, Attention UNet, and LeViT UNet exhibit significant shortcomings in detecting tumor edges and complex regions, which is reflected by a large number of red pixels in their prediction results. This indicates that these models face difficulties in accurately identifying tumor boundaries. UNet, SegNet, and Nested UNet show improvements in reducing false negatives (missed detections), but their false positive (false detections) rates are extremely high. In particular, the Nested UNet model has a substantial number of green pixels in the bottom-right corner, and this issue is largely related to the accuracy of the ground truth labels. In contrast, LiteMRINet demonstrates the best prediction performance: it has almost no false negatives and an extremely low number of false positives. This suggests that LiteMRINet is more sensitive to labels and can distinguish different features based on the annotated regions.

## Discussion

From the experimental results on the two datasets, it is evident that the performance of models is not consistent across different datasets. For example, PSPNet ranked only 6th on the LGG Segmentation Dataset but ranked 3rd on the BraTS2021 Dataset. LeViT performed the worst on the first dataset, with a significant gap from the best model, but it almost performed the best on the second dataset. This indicates that the performance of a model is related to factors such as the annotation accuracy of the dataset and the size of the segmented objects. However, our proposed LiteMRINet model performed the best on both datasets, suggesting that local features play a significant role in the segmentation of small objects.

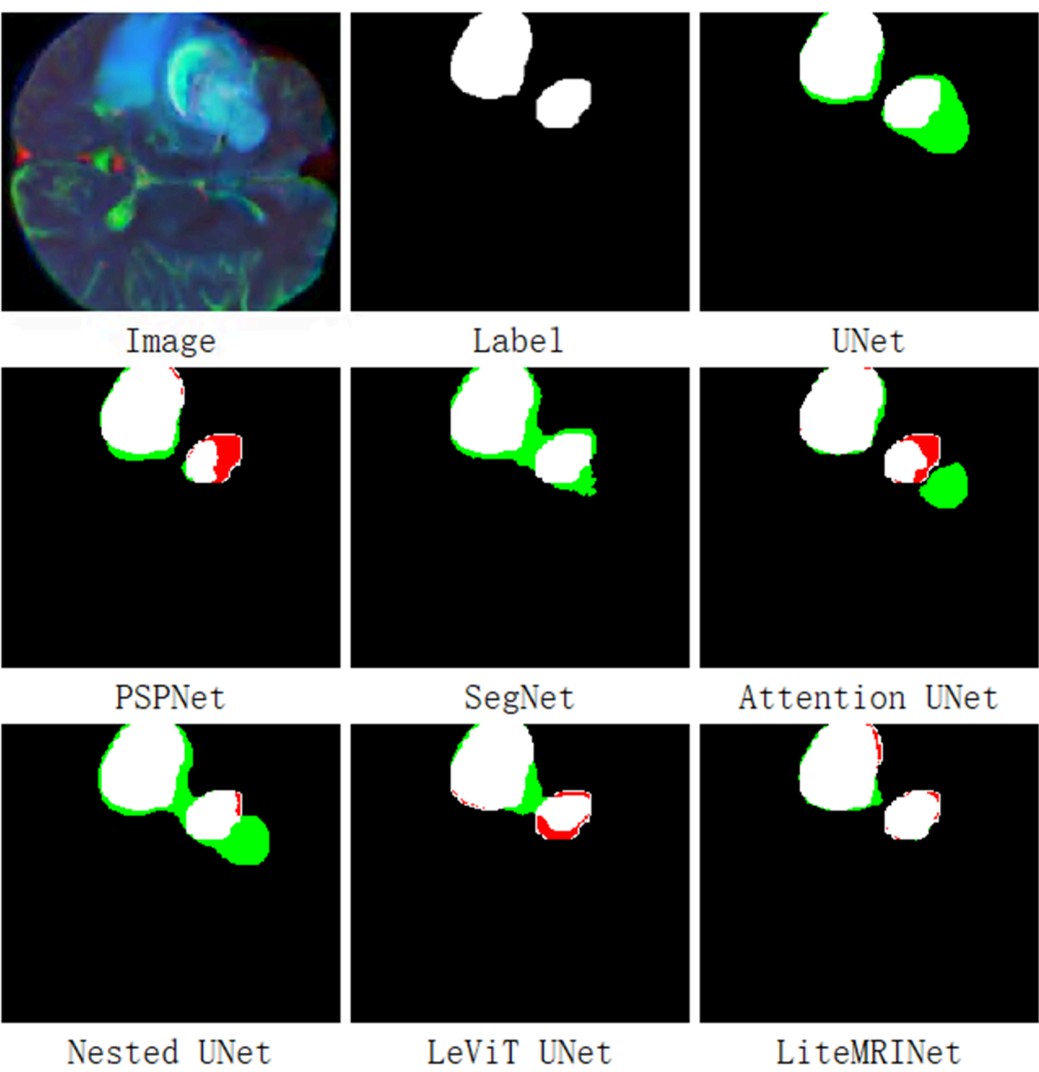

**Fig 6. The output results of each model on the test set.** Black represents TN pixels, white represents TP pixels, red represents FN pixels, and green represents FP pixels.

It is worth noting that the OA values for all models are very close. This is because, in small object segmentation, the background occupies a large proportion of the image. As a result, even if there are false positives or missed detections of tumor pixels, it does not significantly affect the OA.

To study the contribution of each module in LiteMRINet, we conducted ablation experiments on LGG Segmentation Dataset, and the results are shown in Table 3. It can be observed that when using only CNN, the mIoU is only 86.25, F1-Score is 84.2, and the accuracy is 99.7. After replacing the CNN in the last two scales with Transformer, the mIoU, F1-Score, and OA all increased to 86.4, 84.4, and 99.7, respectively. Furthermore, replacing the remaining CNN with three scales of shared CNN further increased all three metrics to 87.2, 85.5, and 99.7, respectively. This demonstrates that the shared CNN module we proposed on three large scales allows each scale to extract deeper features, thereby achieving better segmentation

**Table 3.** Ablation experiment on LGG Segmentation test dataset.

| Methods | mIoU (%) | F1-Score (%) | OA (%) |
|---|---|---|---|
| CNN | $86.25 \pm 0.05$ | $84.2 \pm 0.1$ | $99.7 \pm 0$ |
| Transformer + CNN | $86.4 \pm 0.1$ | $84.4 \pm 0.1$ | $99.7 \pm 0$ |
| Transformer + ShareCNN | $87.2 \pm 0.2$ | $85.5 \pm 0.2$ | $99.7 \pm 0$ |

**Table 4.** Model parameter comparison.

| Methods | Params(M) |
|---|---|
| UNet | 8.64 |
| PSPNet | 46.58 |
| SegNet | 29.44 |
| Attention UNet | 34.88 |
| NestedUNet | 9.16 |
| LeVit Unet | 5.11 |
| LiteMRINet | 1.61 |

results for small targets such as brain tumors. It also demonstrates that using Transformer on a small scale can better capture global features.

To provide a more intuitive comparison of the complexity of the models, we used the 'get_model_complexity_info' function from the 'ptflops' library to obtain the parameter count of networks using different modules, as shown in Table 4.

From the table, we can see that PSPNet has the most parameters, approximately 46.58 million, and we only used ResNet50 as the backbone. Attention UNet and SegNet also have relatively more parameters, with 34.88 million and 29.44 million, respectively. UNet++ and UNet have fewer parameters, with 9.16 million and 8.64 million, respectively. In contrast, our proposed LiteMRINet has significantly fewer parameters than all other networks, with only 1.61 million, which is one-thirtieth of the parameter count of PSPNet, making it advantageous for devices with limited memory.

## Conclusion

To address the contradiction between the need for deeper local features in tumor semantic segmentation in MRI and the requirement for more network parameters for deeper local features, we utilized shared-parameter convolutions on three large-scale networks that capture local features. This approach deepened the network depth for each scale while not increasing the total parameter count of the entire network. Furthermore, to better capture global feature information at small scales, we replaced convolutional operations with Transformers. Since the input feature size at small scales is relatively small, replacing convolutions with Transformers did not significantly increase the number of parameters. With this design, our network can better capture both local and global features, enhancing tumor identification in MRI while significantly reducing the total parameter count of the network. Through analysis, we found that the total parameters of our proposed LiteMRINet model are only about one-fifth of the parameters in UNet and even one-thirtieth of PSPNet's parameters.

On the LGG Segmentation dataset, our proposed network outperformed mainstream semantic segmentation networks in terms of mIoU, F1-score, and OA, achieving scores of 87.2, 85.5, and 99.7, respectively. On the BraTS2021 dataset, LiteMRINet also performs exceptionally well, outperforming other models in terms of both mIoU and F1-score, with mIoU at 91.25% and an F1-score of 91.05%. From the three tumor prediction images, it can be

observed that LiteMRINet had the lowest false detection and miss detection rates, making it closest to the ground truth.

In summary, our LiteMRINet model, compared to mainstream semantic segmentation networks, not only significantly reduces the parameter count but also improves tumor segmentation performance on MRI. This provides a clear advantage, especially on devices with limited memory resources.

## Supporting information

**S1 Dataset. BraTS21 Dataset.**
(ZIP)

**S2 Dataset. LGG Segmentation Dataset.**
(ZIP)

## Acknowledgments

LGG Segmentation Dataset is here, https://www.kaggle.com/datasets/mateuszbuda/lgg-mri-segmentation. BraTS21 Dataset is here https://www.kaggle.com/datasets/rickandjoe/brats21-preprocessed-train-images-with-torchio. Wei Yuan proposed research methods, wrote code, and wrote a manuscript. Han Kang provided research data, reviewed and edited the manuscript. All authors reviewed the manuscript.

## Author contributions

**Data curation:** Han Kang.

**Methodology:** Wei Yuan.

**Software:** Wei Yuan.

**Writing – original draft:** Wei Yuan.

**Writing – review & editing:** Han Kang.

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
