## [Decision Letter · Decision Letter 0]

29 Jul 2025

PONE-D-25-30235Enhanced Local Feature Extraction with Lite Network for Precise Segmentation of Small Brain Tumors in MRIPLOS ONE

Dear Dr. Yuan,

Thank you for submitting your manuscript to PLOS ONE. After careful consideration, we feel that it has merit but does not fully meet PLOS ONE’s publication criteria as it currently stands. Therefore, we invite you to submit a revised version of the manuscript that addresses the points raised during the review process.

We look forward to receiving your revised manuscript.

Kind regards,

Xiaohui Zhang

Academic Editor

PLOS ONE

Journal Requirements:

2. We note that your Data Availability Statement is currently as follows: All relevant data are within the manuscript and in Supporting Information files.

Reviewers' comments:

Reviewer's Responses to Questions

**Comments to the Author**

1. Is the manuscript technically sound, and do the data support the conclusions?

Reviewer #1: Yes

Reviewer #2: Yes

2. Has the statistical analysis been performed appropriately and rigorously?

Reviewer #1: N/A

Reviewer #2: No

3. Have the authors made all data underlying the findings in their manuscript fully available?

Reviewer #1: Yes

Reviewer #2: Yes

4. Is the manuscript presented in an intelligible fashion and written in standard English?

Reviewer #1: Yes

Reviewer #2: No

5. Review Comments to the Author

Reviewer #1: This manuscript presents a lightweight network to perform segmentation task on MRI images utilizing shared CNN layers. The following questions and comments should be addressed during revision:

1. Figure 3, since the authors proposed using shared CNN layers during encoding, why not leverage the parameter saving to explore networks with larger depths? Does increasing the network depth increase the performance of the model?

2. Table 1, the overall accuracy metrics are really close for all models reported. The authors should consider repeating the model training process a few times to obtain the mean and standard deviation of the performance metrics to determine if the improvements are statistically significant.

3. The training time for each model should also be reported.

4. What is the training/testing ratio when the models were trained?

5. Figure 4, the meaning of the red bounding box should be explained in the figure caption.

6. A core assumption of this manuscript is the scale invariance of the CNN layers. Have the authors tested this using the datasets available? For example, for the tumor in Figure 4, what happens to all trained model if we crop the image down to a sub-region containing the tumor (resampling back to the same resolution, of course). This effectively changes the scale of the image/tumor. Is the proposed model able to correctly segment out the tumor in this case?

7. Table 3, for the ablation study, another case with just shared CNN layers (without transformers) should be added to see the impact of just the shared CNN layers.

Reviewer #2: This paper presents LiteMRINet, a lightweight network for segmenting small brain tumors in MRI images. The method introduces a shared 10-layer CNN applied across multiple input scales to enhance local feature extraction while minimizing parameter growth. Additionally, Transformer modules are used on low-resolution feature maps to capture global context. The architecture adopts a U-Net-style decoder and is evaluated on the LGG Segmentation and BraTS21 datasets, demonstrating competitive segmentation performance with significantly fewer parameters compared to existing models. However, before publication, the following concerns should be addressed.

1. Did the study include validation sets? These are typically essential for tuning and early stopping.

2. Does the arthor implement the study to test the generalization ability of proposed method?

3. The usage of “U-Net” and “UNet” should be standardized; “U-Net” is the more widely accepted form.

4. The notation “128×128@3” used to describe data shape is uncommon and should be revised to a more standard format like 128×128×3.

5. The first paragraph of the introduction discusses tumor characteristics and clinical context with minimal referencing. I would suggest add more citations to support those statements.

6. PLOS authors have the option to publish the peer review history of their article (what does this mean?). If published, this will include your full peer review and any attached files.

Reviewer #1: No

Reviewer #2: No

---

## [Author Response · Author response to Decision Letter 1]

28 Aug 2025

Reviewer #1: This manuscript presents a lightweight network to perform segmentation task on MRI images utilizing shared CNN layers. The following questions and comments should be addressed during revision:

1. Figure 3, since the authors proposed using shared CNN layers during encoding, why not leverage the parameter saving to explore networks with larger depths? Does increasing the network depth increase the performance of the model?

Response: Our research objective is to enhance the extraction of large-scale detailed features without increasing the network parameters, and even to reduce the network parameters. Although increasing the number of CNN layers would yield better performance, it would lead to more model parameters and higher requirements for the computer’s GPU.

2. Table 1, the overall accuracy metrics are really close for all models reported. The authors should consider repeating the model training process a few times to obtain the mean and standard deviation of the performance metrics to determine if the improvements are statistically significant.

Response:Thank you for your suggestion. We have replaced the GPU with a 4060Ti 16G,

and conducted multiple rounds of model training and continued until the differences between the three metrics of the two best training results were all within 0.5. We then took the average of these two sets of metrics as the final result. The experimental results indeed differed from the previous ones, so we have rewritten the results section.

3. The training time for each model should also be reported.

Response: Thank you for your suggestion. Our model’s advantage lies in its lightweight nature; however, due to the internal implementation of convolutional operations, a lightweight model does not necessarily ensure faster computation.

4. What is the training/testing ratio when the models were trained?

Response: 3,000 images from the LGG Segmentation Dataset were allocated to the training set, while the remaining 929 images were designated for the test set. We have revised the materials subsection and added the following information: For the BraTS21 dataset, the training set contains 580 images and the test set includes 85 images.

5. Figure 4, the meaning of the red bounding box should be explained in the figure caption.

Response: Rectangular bounding boxes have been removed from the new results.

6. A core assumption of this manuscript is the scale invariance of the CNN layers. Have the authors tested this using the datasets available? For example, for the tumor in Figure 4, what happens to all trained model if we crop the image down to a sub-region containing the tumor (resampling back to the same resolution, of course). This effectively changes the scale of the image/tumor. Is the proposed model able to correctly segment out the tumor in this case?

Response: We enable input images of different scales to share the same convolutional neural network (CNN), thereby training the network to recognize features across multiple scales simultaneously. Our assumption is that tumors of varying sizes should all be identifiable by the same network model—analogous to how the human brain can accurately recognize a car both from a distance and up close.

The dataset itself already contains tumors of different sizes, and all models are capable of recognizing them; the only difference lies in the recognition performance among the models. Due to the enhanced depth of large-scale feature extraction in our network, it achieves better performance while having the fewest parameters among all models.

7. Table 3, for the ablation study, another case with just shared CNN layers (without transformers) should be added to see the impact of just the shared CNN layers.

Response: In fact, the change in metric values from the second row ("Transformer + CNN") to the third row ("Transformer + ShareCNN") directly represents the impact solely brought by the shared CNN layers.

Moreover, since convolution primarily captures local features, we specifically apply ShareCNN to large-scale inputs. If we remove the Transformer and use only ShareCNN (which would mean applying ShareCNN to small-scale inputs as well), this would be of little significance.

Reviewer #2: This paper presents LiteMRINet, a lightweight network for segmenting small brain tumors in MRI images. The method introduces a shared 10-layer CNN applied across multiple input scales to enhance local feature extraction while minimizing parameter growth. Additionally, Transformer modules are used on low-resolution feature maps to capture global context. The architecture adopts a U-Net-style decoder and is evaluated on the LGG Segmentation and BraTS21 datasets, demonstrating competitive segmentation performance with significantly fewer parameters compared to existing models. However, before publication, the following concerns should be addressed.

1. Did the study include validation sets? These are typically essential for tuning and early stopping.

Response: In this experiment, we used a training set and a test set. For early stopping, we adopted the criterion that training is halted if the loss function value does not decrease for 20 consecutive epochs.

2. Does the arthor implement the study to test the generalization ability of proposed method?

Response: In our experimental results, the metric values of all models on the test set represent a generalization outcome of the trained models.

3. The usage of “U-Net” and “UNet” should be standardized; “U-Net” is the more widely accepted form.

Response: Thank you for your suggestion. To avoid having two hyphens ("-") in terms like "LeViT-UNet" and "MA-UNet", we have revised all instances of "U-Net" to "UNet".

4. The notation “128×128@3” used to describe data shape is uncommon and should be revised to a more standard format like 128×128×3.

Response: Thank you for your suggestion; we have revised.

5. The first paragraph of the introduction discusses tumor characteristics and clinical context with minimal referencing. I would suggest add more citations to support those statements.

Response: Thank you for your suggestion; we have revised first two paragraph of introduction.

---

## [Decision Letter · Decision Letter 1]

28 Sep 2025

Enhanced Local Feature Extraction of Lite Network with Scale-Invariant CNN for Precise Segmentation of Small Brain Tumors in MRI

PONE-D-25-30235R1

Dear Dr. Kang,

We’re pleased to inform you that your manuscript has been judged scientifically suitable for publication and will be formally accepted for publication once it meets all outstanding technical requirements.

Kind regards,

Xiaohui Zhang

Academic Editor

PLOS ONE

Additional Editor Comments (optional):

Reviewers' comments:

Reviewer's Responses to Questions

**Comments to the Author**

1. If the authors have adequately addressed your comments raised in a previous round of review and you feel that this manuscript is now acceptable for publication, you may indicate that here to bypass the “Comments to the Author” section, enter your conflict of interest statement in the “Confidential to Editor” section, and submit your "Accept" recommendation.

Reviewer #2: All comments have been addressed

2. Is the manuscript technically sound, and do the data support the conclusions?

Reviewer #2: Yes

3. Has the statistical analysis been performed appropriately and rigorously?

Reviewer #2: Yes

4. Have the authors made all data underlying the findings in their manuscript fully available?

Reviewer #2: Yes

5. Is the manuscript presented in an intelligible fashion and written in standard English?

Reviewer #2: Yes

6. Review Comments to the Author

Reviewer #2: The authors have addressed all my concerns. The paper is technically sound and well-written. Thus, I recommend the manuscript for publication.

7. PLOS authors have the option to publish the peer review history of their article (what does this mean?). If published, this will include your full peer review and any attached files.

Reviewer #2: No

---

## [Editor Report · Acceptance letter]

PONE-D-25-30235R1

PLOS ONE

Dear Dr. Kang,

I'm pleased to inform you that your manuscript has been deemed suitable for publication in PLOS ONE. Congratulations! Your manuscript is now being handed over to our production team.

Kind regards,

on behalf of

Dr. Xiaohui Zhang

Academic Editor

PLOS ONE